# Resistance to Acetylsalicylic Acid in Patients with Coronary Heart Disease Is the Result of Metabolic Activity of Platelets

**DOI:** 10.3390/ph13080178

**Published:** 2020-08-01

**Authors:** Yuriy I. Grinshtein, Andrei A. Savchenko, Aleksandra A. Kosinova, Maxim D. Goncharov

**Affiliations:** 1Therapeutic Department of Institute of Postgraduate Education, Krasnoyarsk State Medical University Named After Prof. V.F. Voyno-Yaseneckiy, 660125 Krasnoyarsk, Russia; grinstein.yi@mail.ru (Y.I.G.); aasavchenko@yandex.ru (A.A.S.); adimax07@mail.ru (M.D.G.); 2Krasnoyarsk Science Center of the Siberian Branch of the Russian Academy of Sciences, Scientific Research Institute of Medical Problems of the North, 660125 Krasnoyarsk, Russia

**Keywords:** acetylsalicylic acid resistance, platelets, Nox2, reactive oxygen species, dehydrogenase

## Abstract

Sensitivity to acetylsalicylic acid (ASA) is important in the treatment of patients with coronary heart disease (CHD) after coronary artery bypass grafting (CABG). Patients were divided into ASA sensitive (sASA) and ASA resistant (rASA) by the activity of platelet aggregation induced arachidonic acid (ARA) together with ASA. Induced platelet aggregation activity was studied in sASA and rASA patients with CHD before and after CABG. The level of synthesis of primary and secondary reactive oxygen species (ROS) by platelets was determined using chemiluminescent analysis. The activity of NAD- and NADP-dependent dehydrogenases in platelets was determined by the bioluminescent method. It was found that the aggregation activity of platelets depended on the sensitivity of CHD patients to ASA and decreased during postoperative ASA therapy. The most pronounced differences in metabolic parameters of platelets in sASA and rASA patients were detected by Nox2 activity. The synthesis of secondary ROS by platelets of CHD patients did not depend on the sensitivity of patients to ASA but increased during postoperative treatment with ASA. The activity of NAD(P)-dependent dehydrogenases in platelets did not differ in sASA and rASA patients with CHD.

## 1. Introduction

Acetylsalicylic acid (ASA) is the most widely used drug for the prevention of cardiovascular events including adverse outcomes in coronary heart disease (CHD). In addition, ASA is used to reduce the risk of shunt occlusion after coronary artery bypass grafting (CABG) [1,2]. ASA acts to impair the development of platelet-mediated atherothromboembolic events by irreversibly inhibition of platelet cyclooxygenase-1 (COX-1) [3,4]. Inhibition of this enzyme prevents the synthesis of potent pro-aggregatory prostanoid thromboxane A2. Nevertheless, ASA does not completely inhibit platelet activation in a significant number of patients. Accordingly, the result of this is the development of vascular thrombosis at the level of the coronary and cerebral vessels as well as the development of arteriovenous and arterial shunt thrombosis after CABG. It is known that from 5% to 60% of CHD patients do not respond to ASA in the secondary prevention of ischemic events [5,6]. This phenomenon was defined as ASA resistance. Single nucleotide polymorphisms, miRNAs, metabolic syndrome, inflammation, malabsorption in the gastrointestinal tract and some others were considered as reasons for the development of ASA resistance [7,8].

We also believe that the functional activity of platelets is largely determined by the state of their metabolism. Accordingly, ASA resistance may also depend on platelet metabolism. It was proven that the functional activation of platelets was carried out with enhanced synthesis of reactive oxygen species (ROS) by enzymes that are located on the outer cytoplasmic membrane and inside the cells (including in mitochondria) [9,10]. The first stage of ROS synthesis is the reaction of the enzymatic NADPH-oxidase complex (NADPH-oxidase, Nox) that is localized mainly on the outer cytoplasmic membrane and carries out the synthesis of superoxide radical (primary ROS). Nox is represented by seven types differing in cell specificity and composition of subunits, Nox2 is detected in platelets [11,12,13]. It was found that low Nox2 activity led to a decrease in platelet aggregation, whereas high Nox2 activity was accompanied by increased platelet aggregation, which in some cases was detected in cardiovascular diseases [9,14]. Superoxide radical dismutation is carried out by superoxide dismutase (SOD). The product of this reaction is hydrogen peroxide (this and subsequent forms are defined as secondary ROS), which can cause oxidation of protein SH-groups and peroxidation of unsaturated fatty acids [15,16,17]. The formation of hydrogen peroxide initiates a cascade of reactions leading to the synthesis of other secondary ROS. Catalase, cyclooxygenase, and glutathione peroxidase also make a significant contribution to the overall level of ROS in platelets [18,19,20]. Primary and secondary ROS are also defined as signaling and regulatory molecules that initiate and modulate the functional activity of platelets [12,15].

Plastic and energy metabolism also contribute to the state of the functional activity of platelets. Plastic reactions determine the synthesis of surface platelet receptors and various humoral factors, while energy metabolism supplies energy for these processes [21,22]. It has been proven that platelet energy is based on glycolysis while preserving mitochondrial functions (aerobic respiration) [23]. In particular, the activity of mitochondrial glycerol-3-phosphate dehydrogenase increases with platelet activation by thrombin [24]. The decrease in platelet count in patients taking ketoprofen was explained by drug inhibition of lactate dehydrogenase activity and, accordingly, a pronounced decrease in the intensity of energy processes in platelets [25].

However, the role of metabolic processes (including synthesis of ROS) in the implementation of the functional activity of platelets in CHD patients with different sensitivity to ASA has not yet been studied. It is important to note that the functional activity of platelets can change after CABG, which is associated with the use of extracorporeal circulation (by activating platelets upon contact with synthetic surfaces) and subsequent postoperative therapy (in particular the use of ASA) of patients [26,27,28]. For example, the article by Kobzar et al. (2020) shows that the level of spontaneous platelet aggregation in patients on days 3 and 5 after CABG was higher than before CABG [27]. Therefore, the study of the properties of platelets in CHD patients should be carried out both before and after CABG, which will allow assessing the change in metabolism and functional activity of platelets for taking measures to reduce the risk of postoperative cardiovascular complications. Thus, the aim of our investigation was to study the metabolic activity of platelets in CHD patients responding and not responding to ASA with a decrease in aggregation.

We chose the activity of NAD- and NADP-dependent dehydrogenases as the studied indicators of platelet metabolism based on the following. Firstly, pyridine nucleotides (coenzymes of dehydrogenases) are the main carriers of electrons in cells and, therefore, these enzymes take an active part in energy processes [29,30]. Secondly, dehydrogenases participate in the directed coordination of conjugated metabolic fluxes and to a large extent determine adaptive changes in cellular metabolism [31,32].

## 2. Results

### 2.1. Platelet Aggregation Activity

The result of platelet aggregation (with various inductors) measured on CHD patients with ASA sensitivity (sASA) and ASA resistance (rASA) before and after CABG are presented in Figure 1. Figure 1a shows the characteristics of ARA-induced platelet aggregation in both groups of CHD patients when ASA was added to the sample with platelets (the method of dividing patients into two groups—sASA and rASA patients). Accordingly, statistically significant differences in platelet aggregation between groups of sASA and rASA patients were observed before CABG (*p* < 0.001) and persisted for one (*p* < 0.001) and 8–10 (*p* = 0.006) days after CABG. The platelet aggregation on sASA patients with CHD before and after CABG was lower than in patients of the control group. Higher level of the platelet aggregation compared with control values was detected on rASA patients before CABG (*p* < 0.001) and the first day after CABG (*p* = 0.042). ARA-Induced platelet aggregation with the addition of ASA on patients of this group on the 8–10th day after CABG corresponded to control values.

There was no significant difference in the ARA-induced (no ASA in samples) platelet aggregation between sASA and rASA patients with CHD before CABG (Figure 1b). Patients with ASA sensitivity had a lower level of ARA-induced platelet aggregation than in control subjects (*p* = 0.010). ARA-Induced platelet aggregation in sASA patients on the 1st day after CABG decreased relative to control (*p* < 0.001) and initial (*p* < 0.001) values and also became lower than in rASA patients (*p* = 0.027). rASA Patients in this period also had a lower level compared with the control (*p* < 0.001) and the initial (*p* = 0.012) values of the ARA-induced platelet aggregation. Similar ratios of ARA-induced platelet aggregation on sASA and rASA patients were also observed eight to 10 days after CABG.

ADP-induced platelet aggregation in sASA and rASA patients with CHD was lower than in control subjects during the entire observation period (*p* < 0.001 with all comparisons) (Figure 1c). There were no significant differences on ADP-induced platelet aggregation between sASA and rASA patients also throughout the examination period. Additionally, it was found that the levels of ADP-induced platelet aggregation in sASA and rASA patients decreased by 8–10 days after CABG relative to initial values (*p* = 0.016 and *p* = 0.002, accordingly).

The level of collagen-induced platelet aggregation in CHD patients before CABG did not differ from control values and there were no differences between sASA and rASA patients (Figure 1d). The level of collagen-induced platelet aggregation in sASA patients was reduced relative to control and initial values on days one (*p* = 0.013 and *p* < 0.001, accordingly) and eight to 10 (*p* < 0.001 in both cases) after CABG. rASA Patients on the first day after the CABG had a reduced level of the aggregation relative to the initial level (*p* = 0.019) whereas on the 8–10th day a decrease was revealed relative to the control and initial values (*p* < 0.001 and *p* = 0.008, accordingly). There were no differences in the levels of collagen-induced platelet aggregation in sASA and rASA patients after CABG.

Levels of adrenaline-induced platelet aggregation in sASA and rASA patients with CHD before and after CABG were reduced relative to control values (Figure 1e). Only sASA patients on the first day after CABG had a decrease in the platelet aggregation relative to initial values (*p* < 0.001). Patients with sensitivity and ASA resistance on the 8-10th day after the CABG also had a decrease in the levels of adrenaline-induced platelet aggregation relative to the initial values (*p* < 0.001 in both cases). Differences in the levels of adrenaline-induced platelet aggregation between sASA and rASA groups of patients with CHD before and after CABG were not detected.

We investigated the features of platelet aggregation in sASA and rASA patients with CHD depending on the type of surgery (on-pump or off-pump CABG) and postoperative therapy (only ASA or ASA + clopidogrel). It was found that the level of ADP-induced platelet aggregation in sASA patients operated on-pump on the 8–10th day after CABG was higher than in off-pump patients: on-pump CABG—Me = 30.5%, C_75_ = 25.0% and C_25_ = 42.0%; off-pump patients—Me = 18.0%, C_75_ = 15.0% and C_25_ = 32.5% (*p* = 0.022). Additionally, the level of ADP-induced platelet aggregation in patients of this group who received ASA and clopidogrel in the postoperative period on days 8–10 after CABG was significantly lower than in patients who received only ASA: ASA + clopidogrel—Me = 15.5%, C_75_ = 12.5% and C_25_ = 21.5%; only ASA—Me = 36.0%, C_75_ = 27.0% and C_25_ = 44.0% (*p* < 0.001). There were no other statistically significant differences in the levels of induced platelet aggregation in sASA and rASA patients with CHD.

### 2.2. Chemiluminescent Platelet Activity

Level of ROS synthesis by platelets was determined using two chemiluminescent indicators—lucigenin and luminol. Lucigenin does not penetrate into the cell and interacts only with the superoxide radical, thereby characterizing the activity of the membrane Nox2 [33,34]. Luminol penetrates the cell and interacts with all types of ROS [35,36]. We investigated the overall level of radical synthesis from the values of Imax and S (area under the chemiluminescence curve), which, respectively, characterizes the maximum synthesis per unit time (sec.) and the total amount of radical (in relative units) during the measurement time (90 min.). The kinetics of ROS synthesis was characterized by Tmax (time to reach the maximum for the chemiluminescent curve). Tmax is determined by the time interval from the moment the cell receives a functional or regulatory effect to the rise of the chemiluminescent curve to a maximum.

The results of a study of the chemiluminescent activity of platelets in CHD patients are presented in Table 1. It was found that spontaneous and ADP-induced lucigenin-enhanced platelet chemiluminescence in sASA patients before and after CABG was significantly higher than in patients of the control group (*p* < 0.05 in all indicators: Tmax, Imax and S). The indicators of lucigenin-enhanced platelet chemiluminescence in rASA patients during the observed period were much less stable. The spontaneous (by indicators Tmax (*p* = 0.012) and Imax (*p* = 0.022)) and ADP-induced (by indicators Imax (*p* = 0.012) and S (*p* = 0.039)) lucigenin-enhanced platelet chemiluminescence in rASA patients before CABG was lower than in sASA patients. The indicators of lucigenin-dependent chemiluminescence in rASA patients during this period corresponded to the control level. Some indicators of this type of chemiluminescent platelet activity in rASA patients on the first day after CABG were also lower than in sASA patients: Tmax (*p* < 0.001) and S (*p* = 0.007) of spontaneous chemiluminescence, Tmax (*p* = 0.004) and Imax of ADP-induced chemiluminescence (*p* = 0.029). rASA Patients in this period had an increase in Imax spontaneous chemiluminescence relative to the control level (*p* = 0.037) as well as a decrease in Tmax relative to initial indicators (*p* = 0.010). The indices of lucigenin-enhanced platelet chemiluminescence in rASA patients 8–10 days after CABG were also lower than in sASA patients: Tmax (*p* < 0.001) and Imax (*p* = 0.021) of spontaneous chemiluminescence, Tmax (*p* = 0.004) and Imax (*p* = 0.005) and S (*p* = 0.014) of ADP-induced chemiluminescence. It was additionally found that the Imax of spontaneous lucigenin-enhanced platelet chemiluminescence in rASA patients during this period was significantly higher than the control (*p* = 0.042) and initial (*p* = 0.041) levels.

Quite other changes in CHD patients were detected by luminol-enhanced platelet chemiluminescence (Table 1). The indices of spontaneous and ADP-induced luminol-enhanced platelet chemiluminescence between patients with sASA and rASA did not differ before CABG. An increase in Imax of spontaneous (*p* = 0.026) and induced (*p* = 0.010) luminol-enhanced chemiluminescence relative to control values was found in sASA patients during this period. A decrease in TmaxT ADP-induced platelet chemiluminescence was also found in rASA patients relative to the control level (*p* = 0.023). Spontaneous and ADP-induced luminol-enhanced platelet chemiluminescence in sASA and rASA patients on the first day after CABG significantly increased relative to control values (in indicators Imax and S). Tmax was the only indicator of luminol-enhanced platelet chemiluminescence in CHD patients, which varies depending on the sensitivity to ASA during this period of the examination (*p* = 0.025). Additionally, it was found that Tmax of spontaneous luminol-enhanced platelet chemiluminescence in rASA patients on the first day after CABG increased relative to the initial levels (*p* = 0.010). Spontaneous and ADP-induced luminol-enhanced chemiluminescence of platelets in sASA patients on the 8–10th day after CABG significantly increased both relative to control values and the initial level (*p* < 0.05 in Imax and S). In addition, sASA patients during this period of the examination had a longer Tmax of spontaneous (*p* = 0.015) and induced (*p* = 0.001) luminol-enhanced platelet chemiluminescence than in people in the control group. Spontaneous luminol-enhanced platelet chemiluminescence in rASA patients 8–10 days after CABG was higher than in control subjects (in Imax *p* = 0.037 and in S *p* = 0.020). In addition, rASA patients during this examination period have a higher value of Imax of ADP-induced luminol-enhanced chemiluminescence of platelets compared with the control (*p* = 0.026) and initial (*p* = 0.023) levels. Statistically important differences depending on acid resistance were found in CHD patients during this observation period for Tmax on spontaneous (*p* = 0.036) and ADP-induced (*p* = 0.021) luminol-enhanced platelet chemiluminescence.

The relationships between platelet aggregation and chemiluminescent activity were investigated using correlation analysis. We did not find statistically significant correlation between these indicators in individuals of the control group. It was found that the level of ADP-induced aggregation was negatively correlated with Imax ADP-induced lucigenin- and luminol-enhanced platelet chemiluminescence in sASA patients with CHD before CABG (*r* = −0.27, *p* = 0.048 and *r* = −0.28, *p* = 0.042, accordingly). The level of adrenaline-induced aggregation also negatively correlated with the Imax and S of spontaneous lucigenin-enhanced chemiluminescence of platelets in sASA patients during this observation period (*r* = −0.31, *p* = 0.024 and *r* = −0.30, *p* = 0.030, accordingly). In addition, the level of ARA-induced platelet aggregation with the addition of ASA in sASA patients before CABG was negatively correlated with the Imax of ADP-induced lucigenin-enhanced (*r* = −0.27, *p* = 0.046) and S of spontaneous luminol-enhanced (*r* = −0.31, *p* = 0.022) platelet chemiluminescence. We did not reveal the relationship between the aggregation and chemiluminescent activity of platelets in sASA patients after CABG. Patients with rASA before CABG had positive correlations of ARA-induced aggregation with S of spontaneous (*r* = 0.63, *p* = 0.029) as well as Imax (*r* = 0.85, *p* < 0.001) and S (*r* = 0.87, *p* < 0.001) of ADP-induced luminol-enhanced chemiluminescence of platelets. The level of ADP-induced aggregation in rASA patients on the first day after CABG was also positively correlated with the Imax of spontaneous (*r* = 0.81, *p* = 0.002) and ADP-induced (*r* = 0.68, *p* = 0.016) luminol-enhanced platelet chemiluminescence. The level of ARA-induced aggregation in patients of this group on the 8–10th day after CABG was interrelated with the Imax of spontaneous (*r* = 0.76, *p* = 0.004) and ADP-induced (*r* = 0.59, *p* = 0.043) luminol-enhanced platelet chemiluminescence.

We did not find statistically significant differences in the parameters of chemiluminescent activity in sASA and rASA patients with CHD depending on on-pump or off-pump CABG and postoperative therapy (only ASA or ASA + clopidogrel).

### 2.3. The Activity of NAD- and NADP-Dependent Dehydrogenases in Platelets

We investigated the activity of NAD- and NADP-dependent dehydrogenases in sASA and rASA platelets with CHD in patients before and after CABG. It was found that the activity of glucose-6-phosphate dehydrogenase (Glu6PDH) in the platelets of CHD patients before and after CABG was significantly lower than in individuals in the control group (*p* < 0.05 in all cases) (Figure 2a). The activity of the aerobic reaction of lactate dehydrogenase (LDH) in sASA and rASA patients was reduced relative to control values before CABG (*p* = 0.002 and *p* = 0.004, accordingly) and on the first day after CABG (*p* = 0.002 and *p* = 0.029, accordingly) (Figure 2b). The activity of this enzyme was reduced relative to the control (*p* < 0.001) and initial (*p* = 0.008) levels in platelets in sASA patients at 8–10 days after CABG, while the enzyme activity in rASA patients during this observation period corresponded to control values. The activity of the anaerobic reaction of lactate dehydrogenase (NADH-LDH) in platelets is significantly increased in sASA patients with CHD before CABG (*p* = 0.002) and also on the first (*p* = 0.002) and 8–10th day (*p* = 0.004) after CABG relative to control values (Figure 2c). The activity of this enzyme in platelets in rASA patients was lower than in individuals in the control group only before CABG (*p* = 0.047). The activity of the NADH-dependent glutamate dehydrogenase reaction (NADH-GluDH) in platelets decreased relative to the initial level on the first (*p* = 0.042) and 8–10th (*p* < 0.001) days after CABG only in sASA patients (Figure 2d). No statistically significant changes in the activity of the remaining dehydrogenases in platelets were detected in sASA and rASA patients with CHD.

The activity of enzymes in platelets in sASA and rASA patients with CHD did not differ significantly depending on the use of on-pump CABG and the type of postoperative treatment (only ASA or ASA + clopidogrel).

The relationships between platelet aggregation ability and enzyme activity were investigated using correlation analysis. People in the control group did not have statistically significant correlations between these indicators. The level of ADP-induced platelet aggregation was positively correlated with NADPH-GluDH activity in sASA patients before CABG (*r* = 0.33, *p* = 0.014), while this type of aggregation was negatively correlated with NADP-dependent isocitrate dehydrogenase (NADP-ICDH) activity in rASA patients (*r* = −0.63, *p* = 0.027) during this examination period. The level of ARA-induced platelet aggregation in sASA patients on the first day after CABG was positively correlated with the activity of the NADH-GluDH (*r* = 0.41, *p* = 0.002), while the level of ARA-induced aggregation with the addition of ASA already negatively correlated with the activity of the glucose-6-phosphate dehydrogenase (Glu6PDH) in these patients (*r* = −0.28, *p* = 0.044). Glutathione reductase (GR) activity was positively correlated with the levels of ADP- and ARA-induced platelet aggregation (*r* = 0.71, *p* = 0.010 and *r* = 0.75, *p* = 0.005, accordingly) in rASA patients on the 1st day after CABG, while NADH-GluDH activity was already negatively correlated with these types of platelet aggregation activity (*r* = −0.63, *p* = 0.027 and *r* = −0.62, *p* = 0.033, accordingly). The NADH-LDH activity was positively correlated with the level of ARA-induced platelet aggregation in sASA patients (*r* = 0.28, *p* = 0.041) on the 8–10th day after CABG, and this enzyme was negatively correlated with the level of ADP-induced platelet aggregation in rASA patients (*r* = −0.84, *p* = 0.001) during this observation period.

## 3. Discussion

In general, platelet aggregation activity in patients with CHD was lower than in people of the control group and depended on the sensitivity of patients to ASA, conceivably due to medications that are taken by patients before CABG. However, some features were discovered. The activity of the ARA-induced platelet aggregation together with the ASA in rASA patients after the operation consistently decreased. This result reflects a decrease in the number of ASA resistant patients in this group during post-operative ASA therapy. At the same time, the most pronounced differences in platelet aggregation activity depending on patients’ resistance to ASA were found by the example of ARA-induced aggregation: the level of aggregation in rASA patients after CABG was higher than in sASA patients. Platelet aggregation induced by ADP, collagen and adrenaline in CHD patients during postoperative ASA therapy was reduced and did not depend on the sensitivity of patients to ASA.

We investigated the levels of synthesis of ROS by platelets in sASA and rASA patients with CHD before and after CABG using chemiluminescent analysis. Lucigenin-enhanced chemiluminescence characterizes the activity of the Nox2 and, accordingly, the level of superoxide radical synthesis [33,34]. We found that the activity of the Nox2 in the platelets of sASA patients before and after CABG was significantly higher than in individuals in the control group. The overall level of synthesis and kinetics of the synthesis of superoxide radical in patients of this group practically did not change in the dynamics of observation. Platelets of sASA patients with CHD synthesized superoxide radical more actively, but the activation time of Nox2 was extended. The activation of the Nox2 was detected when the platelets were in a state of relative rest and during their functional activation. Additionally, the activation time of the enzyme was longer when the platelets were at rest and during their functional activation. The activity of Nox2 in platelets in rASA patients before and after CABG was lower than in sASA patients. However, the kinetics of the synthesis of superoxide radical by platelets in patients of this group was significantly different from that found in sASA patients. First, the rate of Nox2 activation of platelets in a state of relative rest (spontaneous lucigenin-enhanced chemiluminescence) in rASA patients before and after CABG was lower than in sASA patients and corresponded to control values. Secondly, the rate of increase in the synthesis of superoxide radical in ADP-dependent platelet activation in rASA patients corresponded to the levels detected in the control group and patients and on the first day after the operation was even larger. Therefore, the activity of Nox2 in rASA platelets in patients is formed by two mechanisms: membrane expression level (lower than in sASA patients) and regulatory processes that are leveled upon cell activation. It was previously shown that low Nox2 activity caused a decrease in the functional activity of platelets [9,10,11,12,13]. We found using correlation analysis that platelet aggregation activity was correlated with platelet Nox2 activity only in sASA patients and only before CABG. Therefore, the absence of correlations between platelet aggregation activity and Nox2 activity was determined by ASA resistance and postoperative ASA therapy. Inhibition of Nox2 activity by ASA was shown in an article by Wang et al. [37].

The level of synthesis of secondary ROS was investigated using luminol-enhanced chemiluminescence [36,37]. We did not find statistically significant differences in the levels of synthesis of secondary ROS (luminol-enhanced chemiluminescence) in platelets in sASA and rASA patients with CHD before and after CABG. The level of synthesis of secondary ROS in platelets in sASA patients was increased relative to the control values before and after CABG. The synthesis of secondary ROS in platelets of rASA patients was increased relative to control values only after CABG. The relationship between aggregation activity and the level of synthesis of secondary ROS of platelets in sASA patients were found only before CABG. Moreover, these relationships were negative. At the same time, these relationships in rASA patients were detected both before and after CABG and they were only positive. It has been found that acid can stimulate the synthesis of secondary ROS by platelets [37]. Our study showed that a more pronounced increase in the synthesis of secondary ROS by platelets was observed during ASA therapy (after CABG) and did not depend on the sensitivity of CHD patients to ASA.

The functional activity of platelets is also determined by the activity of NAD- and NADP-dependent dehydrogenases [25,38]. We found that the activity of Glu6PDH in platelets of CHD patients was reduced relative to the control values before and after Glu6PDH and did not depend on the sensitivity of patients to ASA. This enzyme is key and initializing in the pentose phosphate cycle and its role in cell metabolism is of increased interest including for patients with cardiovascular diseases [39]. LDH was also reduced in the platelets of patients with the exception of sASA and rASA patients with CHD on the 8–10th day after CABG. This enzymatic reaction determines the conversion of lactate to pyruvate, which can be used for aerobic processes. The activity of NADH-LDH in the platelets of sASA and rASA patients was increased relative to the control values before CABG. At the same time, the activity of this enzymatic reaction remained elevated after CABG only in sASA patients. The activity of NADH-LDH characterizes the intensity of anaerobic glycolysis that determines a decrease in the value of anaerobic energy for platelets of rASA patients with ASA therapy. We also found that sASA patients on the 8–10th day after CABG had a positive relationship between the activity of this enzyme and platelet aggregation activity, while rASA patients during the same observation period had a negative relationship. A change in the activity of NADH-GluDH relative to initial values was found only in sASA patients. Moreover, the identified relationships between platelet aggregation activity and NADH-GluDH activity in CHD patients differed: with ASA sensitivity—positive, with ASA resistance—negative. This enzymatic reaction is involved in nitrogen exchange reactions in cells and carries out the transfer of substrates from amino acid exchange reactions to tricarboxylic acid cycle reactions [40,41].

It can be concluded that changes in platelet metabolism in CHD patients depending on sensitivity to ASA are more related to enzymatic reactions that determine the activity of energy processes (both anaerobic and aerobic). In general, platelet activation is significantly dependent on the intensity of the substrate flow through glycolysis [42]. Moreover, the activity of aldolase was reduced in platelets of patients with ASA resistance, while a compensatory increase in the activity of glyceraldehyde-3-phosphate dehydrogenase was revealed that was realized in the normal level of pyruvate accumulated in anaerobic glycolysis [43]. Plastic and antioxidant processes are also important for maintaining the vital functions and functional activity of platelets. In particular, it was shown that a low level of antioxidant reactions could not provide adequate protection of platelets in rASA patients from ROS [44]. The mechanism for the development of ASA resistance proposed by Thomas et al. (2014) was in the outflow of metabolites for the synthesis of prostaglandins and, accordingly, in a decrease in the activity of oxidative processes in platelets [44]. However, the mechanisms of the influence of enzyme activity on the functional activity of cells can be determined not only by their role in synthetic and energy processes. The influence of certain metabolic intermediates on epigenetic processes determines the importance of metabolism on the long-term programming of the functional activity of cells. Glutamate, glutathione succinate and some other intermediates have been identified as similar regulators of epigenetic processes [45,46].

It should be noted that understanding the metabolic mechanisms of ASA resistance is also important for several other diseases, in particular type 2 diabetes mellitus (T2DM). It is known that T2DM is accompanied by an increased risk of thrombosis due to increased functional activity of platelets [47,48,49]. In turn, the high functional activity of platelets in this disease was determined by increased sensitivity to aggregation inducers (ADP, thrombin, collagen, etc.), but also metabolic disorders (violation of carbohydrate and lipid metabolism) [50,51,52]. Accordingly, ASA resistance in T2DM patients is of considerable concern, as it reduces the effectiveness of prevention of cardiovascular complications in them [53,54,55]. Therefore, the determination of metabolic mechanisms of platelet resistance to ASA will make it possible to determine the targets of drug exposure to restore sensitivity to ASA and increase the effectiveness of prevention of cardiovascular complications.

Thus, the aggregation activity of platelets depended on the sensitivity of CHD patients to ASA and decreased during postoperative ASA therapy. The activity of ARA-induced platelet aggregation most differentially differentiated CHD patients depending on their sensitivity to ASA. Platelet aggregation activity induced by ADP, collagen and adrenaline did not differ in patients, but decreased during postoperative ASA treatment. The state of platelet metabolism in sASA and rASA patients with CHD varied both before and after CABG. The most pronounced differences in metabolic parameters of platelets in sASA and rASA patients were detected by Nox2 activity. rASA patients had low oxidase activity compared with sASA patients. The synthesis of secondary ROS by platelets of CHD patients did not depend on the sensitivity of patients to ASA but increased during postoperative treatment with ASA. The activity of NAD(P)-dependent dehydrogenases in platelets did not differ in sASA and rASA patients with CHD. However, the activity of dehydrogenases in platelets in sASA patients was more sensitive to postoperative ASA therapy. Moreover, dehydrogenases involved in energy processes were more sensitive to the effects of ASA. Due to the fact that metabolic intermediates (ROS and other products of enzymatic reactions) not only have a metabolic effect but also affect regulatory and epigenetic processes, we can conclude that sensitivity to ASA can be determined by platelet metabolism.

## 4. Materials and Methods

### 4.1. Patients

Sixty-six patients with stable angina pectoris, grade II–III according to the Canadian Cardiovascular Society classification were enrolled in the study (49 men and 17 women) (Table 2). The control group included 16 healthy volunteers (10 men and 6 women, mean age 45.2 ± 9.8 years). All patients underwent CABG; 53 patients (80.3%) underwent on-pump CABG, and 13 patients (19.7%) underwent off-pump CABG. Inclusion criteria were: stable angina pectoris grades II–IV; coronary artery atherosclerosis, proved by coronary angiography; written informed consent. Exclusion criteria were: renal failure; hepatic failure; peptic ulcer and/or 12 duodenal ulcers in the acute stage; ASA or clopidogrel intolerance.

The study protocol was approved by the Ethics Committee of the Krasnoyarsk State Medical University (protocol №76/2016 from 04/05/2017). It was conducted in accordance with the Declaration of Helsinki and was consistent with applicable guidelines for good clinical practice. Written consent was obtained from all participants. During the hospitalization, all patients received therapy according to the Russian Cardiology Society guidelines. The antiplatelet therapy was stopped for at least 5 days before CABG. In the post-surgical period, enrolled patients were treated with 100 mg of enteric form ASA along (39 patients) either 100 mg ASA + clopidogrel 75 mg per day (27 patients). In all patients ASA was started on the first day after CABG. Patients took following medications before CABG: β-blockers—92.4% of patients, Angiotensin-converting enzyme inhibitors—90.9%, Angiotensinogen II receptor blockers—5%, Calcium channel blockers—16.9%, Diuretics—42.4%, Aldosterone antagonists—40.6%, Statins—100%. Median of heparin dose in the group of patients during on-pump CABG 240 mg (220–300), during off-pump CABG—370 mg (320–445), *p* < 0.001.

Forty-three (65.2%) patients had stable angina grade II, 23 (34.8%) patients had grade III of angina pectoris. The age was from 36 to 72 years (mean age 60.9 ± 6.9 years). All patients had atherosclerotic lesions of coronary arteries, proven by coronaroangiography. The baseline characteristic is shown in Table 2.

The venous blood samples were acquired from patients before coronary artery bypass grafting (CABG) after five days of ASA withdrawal, on the first day after surgery and on the 8–10th day after surgery. We have selected these time points to estimate platelet function and metabolism in stable condition before CABG, on the first day after on-pump or off-pump CABG to realize its influence on platelets and on the 8–10th day after surgery as the last day of treatment in a hospital before discharge.

Twelve patients (18.2%) were resistant to ASA.

### 4.2. Determination of ASA Resistance

Blood samples were collected from the antecubital vein in vacutainer tube with 3.8% tri-sodium citrate, in the morning and on an empty stomach. Blood was centrifuged for 10 min at 140 g at room temperature to produce platelet-rich plasma. The supernatant was carefully transferred into a plastic tube and adjusted to 300 × 10^9^ platelets/L with buffer (90 mM NaCl, 5 mM KCl, 36 mM sodium citrate, 10 mM Na_2_-EDTA, pH 7.2). Evaluation of the number of the isolated platelets was carried out on a hematological analyzer Sysmex XE-5000 (Sysmex Inc., Mundelein, IL, USA) by a standard method [56]. Platelet aggregation induced by 0.5 mM arachidonic acid (ARA) was studied on an optical aggregometer 490 Chrono-Log (Havertown, PA, USA) with a self-calibration system. Another sample of platelet-rich plasma from this patient was incubated with 3.36 mM ASA for 3 min and then ARA-induced aggregation of platelets was also studied. The value of the ASA inhibition coefficient of platelet aggregation was calculated by the difference between the levels of aggregation without and with ASA. A high level of the coefficient testified to the sensitivity of the patient to ASA, a low value of the coefficient indicates the patient’s resistance to ASA. The ASA concentration of 3.36 mM was chosen by us during preliminary titration in the range from 0.5 mM to 5 mM with subsequent curve fitting and selection of the maximum.

### 4.3. Platelet Aggregometry Assay

Blood with 3.8% tri-sodium citrate (9:1 *v*/*v*) was centrifuged for 10 min at 140× *g* at room temperature to produce the platelet-rich plasma. The platelet-poor plasma was obtained by further centrifuging the remainder of the platelet-rich plasma specimen at 700× *g* for 20 min at room temperature. Platelet aggregation was determined on an optical aggregometer 490 Chrono-Log (USA) with a self-calibration system. 500 μL platelet-poor plasma was placed in an aggregometer comparison cuvette. Next, 500 μL of platelet-rich plasma was added to a cuvette of an aggregometer with a magnetic stirrer. Samples were incubated for 2 min at a temperature of 37 °C. Then, the aggregation inducer was added to the platelet-rich plasma cuvette and the level of aggregation was measured relative to the cuvette with platelet-poor plasma for 5–6 min. Collagen (2 μg/mL), ADP (5 μM), adrenaline (10 μM) and ARA (0.5 mM) were used as platelet aggregation inducers. Platelet aggregation was expressed as a percentage.

### 4.4. Platelet Isolation

Platelet-rich plasma was obtained by centrifuging stabilized blood (sodium citrate) at 140× *g* for 10 min. The supernatant was carefully collected, transferred to a plastic tube and adjusted to 10 mL with buffer A (90 mM NaCl, 5 mM KCl, 36 mM sodium citrate, 10 mM Na_2_-EDTA, pH 7.2). The resulting cell suspension was centrifuged for 15 min at 400× *g*. The cell pellet was resuspended in 10 mL of buffer A and centrifuged again for 1 min at 400× *g*. Then, 9 mL of the supernatant was collected and centrifuged again for 15 min at 400× *g*. The supernatant was carefully removed and 10 mL of buffer B (130 mM NaCl, 20 mM Tris-HCl buffer, 30 mM Na_2_-EDTA, 15 mM glucose, pH 7.4) was added to the pellet. The cell suspension was centrifuged in the same mode. The supernatant was removed and the pellet was diluted in 400 μL of buffer B and centrifuged for 1 min at 140× *g*. Platelet-containing supernatant was transferred to a plastic tube and used for further investigations. Evaluation of the number and purity of the isolated platelets was carried out on a hematological analyzer Sysmex XE-5000 (Sysmex Inc., Mundelein, IL, USA) by a standard method [56]. The purity of the isolated platelets was 98–100%.

### 4.5. Chemiluminescent Analysis

10^8^ platelets per sample were used to conduct a chemiluminescent reaction. The composition of the reaction sample to determine spontaneous platelet chemiluminescence also included 50 μL of lucigenin (50 μg/mL) or luminol (50 μg/mL) and 250 μL of buffer B (130 mM NaCl, 20 mM Tris-HCl buffer, 30 mM Na_2_-EDTA, 15 mM glucose, pH 7.4). The sample composition for determining ADP-induced platelet chemiluminescence included platelets and chemiluminescent indicators in the same amount, 200 μL of buffer B and 50 μL of 0.1 M ADP. Evaluation of spontaneous and ADP-induced platelet chemiluminescence was performed for 90 min on the 36-channel biochemiluminescence analyzer BLM-3607 (MedBioTech Ltd., Krasnoyarsk, Russia). The following indicators of the chemiluminescent platelet reaction were determined: time to maximum (Tmax), maximum intensity (Imax) and the area (S) under the chemiluminescence curve. The enhancement of the chemiluminescence of induced ADP (activation index—AI) was determined by the ratio of the area under the curve of ADP-induced chemiluminescence to the area under the curve of spontaneous chemiluminescence.

### 4.6. Bioluminescent Analysis

The activity of NAD- and NADP-dependent dehydrogenases in platelets was determined using the bioluminescent method [57]. The activity of the following dehydrogenases was determined:−glucose-6-phosphate dehydrogenase (Glu6PDH);−glycerol-3-phosphate dehydrogenase (Gly3PDH);−NADP-dependent malate dehydrogenase decarboxylated (NADP-MDH);−lactate dehydrogenase—according to NAD- and NADH-dependent reactions (respectively, LDH and NADH-LDH);−malate dehydrogenase—according to NAD- and NADH-dependent reactions (respectively, MDH and NADH-MDH);−NADP-dependent glutamate dehydrogenase—according to NADP- and NADPH-dependent reactions (respectively, NADP-GluDH and NADPH-GluDH);−NAD-dependent glutamate dehydrogenase—according to NAD- and NADH-dependent reactions (respectively, NAD-GluDH and NADH-GluDH);−NADP-dependent isocitrate dehydrogenase (NADP-ICDH);−NAD-dependent isocitrate dehydrogenase (NAD-ICDH);−glutathione reductase (GR).

The platelets were previously destroyed by osmotic lysis (1:5 by volume) with addition of 2 mM dithiotreitol (Sigma-Aldrich, St. Louis, MO, USA). Bioluminescent analysis was performed using NAD(P):FMN oxidoreductase-luciferase (EC 1.5.1.29 and EC 1.14.14.3, respectively) from *Photobacterium leiognathi* (Institute of Biophysics, Siberian Division of Russian Academy of Sciences, Krasnoyarsk) [58]. Activities of the NAD(P)-dependent dehydrogenases in platelets were determined on the 36-channel biochemiluminescence analyzer BLM-3607 (MedBioTech Ltd., Krasnoyarsk, Russia) and were expressed in enzymatic units on 1 mg of protein (1 U = 1 μmol/min) [32]. Protein concentration was determined using a test kit Pierce™ Modified Lowry Protein Assay Kit (Thermo Fisher Scientific, Waltham, MA, USA).

### 4.7. Statistical Analysis

Statistical description was performed by counting the median (Me) and the interquartile ranges in the form of 25 and 75 percentiles (C_25_—C_75_). Significance of differences between indicators was assessed by nonparametric criterion Mann-Whitney U test. The statistical significance of the differences in the dynamics of treatment (dependent variables) was determined by the Wilcoxon matched pairs test. To compare categorical variables, the χ2 test was used, except when expected frequencies in contingency tables were less than 5, in which case the Fisher exact test was used. Spearman rank correlation coefficients were calculated to characterize the strength of the relationship between platelet aggregation activity and metabolic parameters. Statistical analysis was performed in an application package Statistica 6.1 (StatSoft Inc., Tulsa, OK, USA).

## Figures and Tables

**Figure 1 pharmaceuticals-13-00178-f001:**
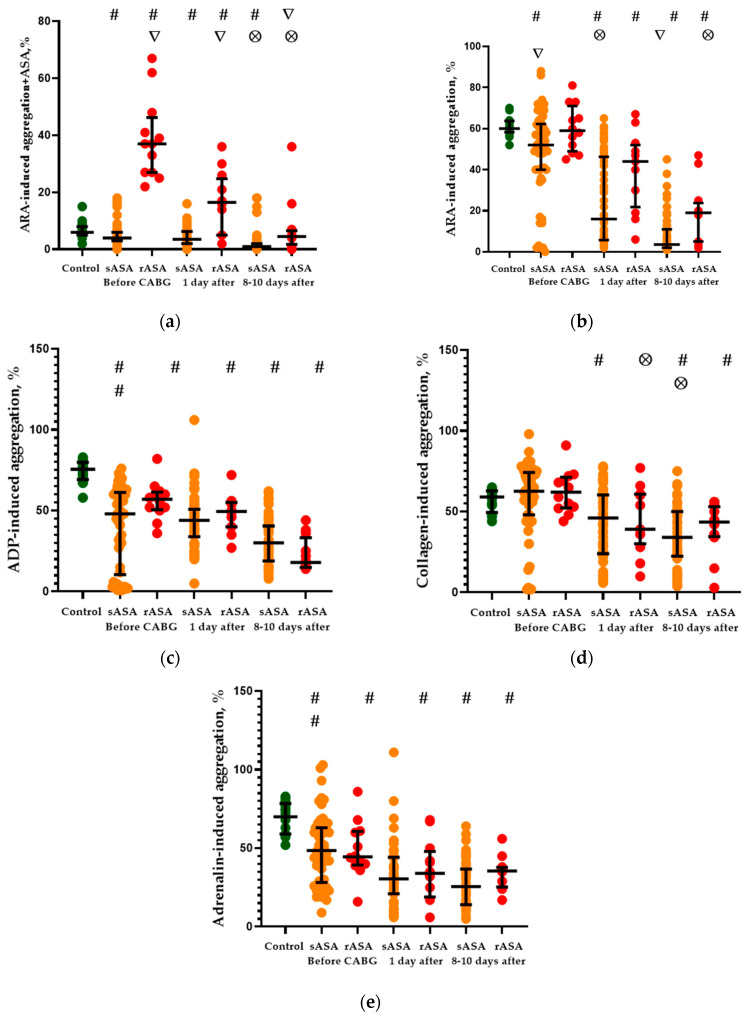
Induced platelet aggregation in sASA and rASA patients with CHD before and after CABG. (**a**) ARA-induced platelet aggregation with the addition of ASA. (**b**) ARA-induced platelet aggregation. (**c**) ADP-induced platelet aggregation. (**d**) Collagen-induced platelet aggregation. (**e**) Adrenalin-induced platelet aggregation. The data represent the medians and interquartile ranges (Me (C_25_—C_75_)). **#**: *p* < 0.05 vs. control (Mann-Whitney U test), **∇**: *p* < 0.05 between indicators of sASA and rASA patients in each period of the survey (Mann-Whitney U test), **⊗**: *p* < 0.05 vs. with indicators of the patients before CABG (Wilcoxon matched pairs test).

**Figure 2 pharmaceuticals-13-00178-f002:**
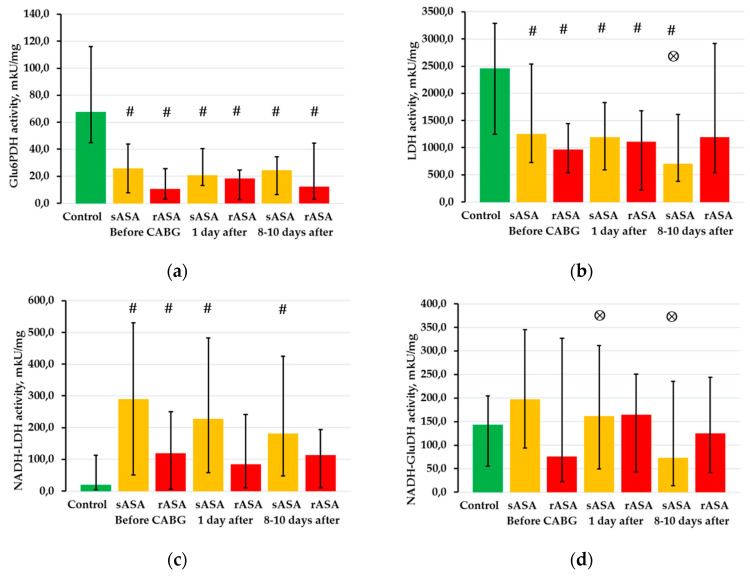
The activity of NAD(P)-dependent dehydrogenases in sASA and rASA patients with CHD before and after CABG. (**a**) The activity of glucose-6-phosphate dehydrogenase. (**b**) The activity of the NAD-dependent reaction of lactate dehydrogenase (aerobic reaction). (**c**) The activity of the NADH-dependent reaction of lactate dehydrogenase (anaerobic reaction). (**d**) The activity of the NADH-dependent reaction of glutamate dehydrogenase. The data represent the medians and interquartile ranges (Me (C25—C75)). **#**: *p* < 0.05 vs. control (Mann-Whitney *U* test), **⊗**: *p* < 0.05 vs. with indicators of the patients before CABG (Wilcoxon matched pairs test).

**Table 1 pharmaceuticals-13-00178-t001:** Chemiluminescent activity of platelet in patients with CHD before and after CABG.

Parameters	Control 1	Before CABG		1 Day After CABG		8–10 days after CABG	
		sASA Patients 2	rASA Patients 3	sASA Patients 4	rASA Patients 5	sASA Patients 6	rASA Patients 7
Spontaneous lucigenin-enhanced chemiluminescence							
Tmax, sec.	213 (80–450)	813 (88–2841) ^#^	185 (35–249) ^∇^	789 (283–2043) ^#^	66 (41–115) ^∇^	938 (565–1908) ^#^	88 (65–420) ^∇^
Imax, r.u.	80 (73–93)	117 (78–566) ^#^	80 (72–92) ^∇^	105 (87–331) ^#^	94 (90–103) ^#^	173 (100–351) ^#^	106 (78–118) ^#^^,^^∇^^,^^⊗^
S, r.u.× sec. × 10^2^	2.38 (1.76–2.75)	3.01 (1.79–7.74) ^#^	2.25 (1.95–3.13)	4.05 (2.44–8.17) ^#^	2.38 (1.97–3.58) ^∇^	4.24 (2.75–6.58) ^#^	2.89 (2.22–3.66)
ADP-induced lucigenin-enhanced chemiluminescence							
Tmax, sec.	96 (49–608)	1036 (346–3743) ^#^	577 (342–1166)	745 (355–1008) ^#^^,^^⊗^	285 (217–341) ^∇^^,^^⊗^	1266 (621–2198) ^#^	495 (263–1099)
Imax, r.u.	80 (76–127)	127 (84–498) ^#^	81 (73–109) ^∇^	360 (91–595) ^#^	98 (82–106) ^∇^	238 (119–455) ^#^	102 (81–129) ^∇^
S, r.u. × sec. × 10^2^	2,75 (1.87–3.65)	4.15 (2.51–10.89) ^#^	2.33 (2.06–3.32) ^∇^	4.86 (2.20–9.50) ^#^	3.05 (2.54–3.48)	4.49 (3.35–7.31) ^#^	2.99 (2.49–3.80) ^∇^
AI	1.01 (0.86–1.87)	1.12 (0.90–1.59)	1.06 (0.92–1.40)	1.11 (0.82–1.40)	1.22 (0.91–1.37)	1.22 (0.99–1.54)	1.09 (1.02–1.17)
Spontaneous luminol-enhanced chemiluminescence							
Tmax, sec.	71 (0–464)	230 (45–1748)	71 (69–81)	336 (71–998)	852 (26–2394) ^⊗^	269 (71–848) ^#^	54 (4–445) ^∇^
Imax, r.u.	80 (77–110)	122 (80–611) ^#^	84 (80–381)	205 (95–490) ^#^	137 (90–167) ^#^	561 (127–1116) ^#^^,^^⊗^	165 (129–388) ^#^
S, r.u. × sec. × 10^2^	2.62 (2.22–3.13)	2.96 (2.10–8.99)	3.01 (2.31–3.41)	3.87 (2.71–9.70) ^#^	3.94 (3.20–4.75) ^#^	5.60 (3.96–12.35) ^#^^,^^⊗^	3.63 (3.14–5.08) ^#^^,^^⊗^
ADP-induced luminol-enhanced chemiluminescence							
Tmax, sec.	154 (0–471)	455 (45–2197)	68 (13–743) ^#^	634 (89–1567) ^#^	97 (69–241) ^∇^	631 (264–1483) ^#^	117 (28–530) ^∇^
Imax, r.u.	77 (71–100)	113 (79–519) ^#^	90 (75–342)	294 (96–785) ^#^	145 (88–188) ^#^	690 (156–1346) ^#^^,^^⊗^	156 (120–398) ^#^^,^^⊗^
S, r.u. × sec. × 10^2^	2.35 (2.07–3.34)	3.06 (2.24–8.35)	3,27 (2.48–3.58)	5.55 (2.37–11.51) ^#^	3.62 (3.20–5.25) ^#^	6.87 (3.87–20.11) ^#^^,^^⊗^	4.74 (2.60–5.92)
AI	0.99 (0.71–1.26)	1.04 (0.73–1.45)	1.10 (1.04–1.39)	1.09 (0.84–1.32)	1.00 (0.69–1.13)	1.16 (0.85–1.37)	1.06 (0.78–1.28)

The data represent the medians and interquartile ranges (Me (C_25_—C_75_)). **^#^**: *p* < 0.05 vs. control (Mann-Whitney *U* test), ^∇^: *p* < 0.05 between indicators of sASA and rASA patients in each period of the survey (Mann-Whitney *U* test), ^⊗^: *p* < 0.05 vs. with indicators of the patients before CABG (Wilcoxon matched pairs test).

**Table 2 pharmaceuticals-13-00178-t002:** Baseline characteristics of participants completing the study.

Parameters	All Patients	sASA Patients	rASA Patients	*p*
Gender, *n* (%) Female/male	17 (25.7%)/ 49 (74.3%)	3 (26.0%)/ 9 (74.0%)	14 (25.0%)/ 40 (75.0%)	0.661
Age (years), Me (C_25—_C_75_)	62.0 (60.5–62.0)	62.5 (56.0–65.0)	63.0 (55.0–65.3)	0.716
Smokers (current), %	36.4	35.8	38.4	0.575
Total cholesterol, mmol/L	4.49 (3.8–5.86)	4.7 (3.8–5.9)	4.2 (3.8–5.8)	0.912
Leukocytes, 10^9^/L	7.66 (6.66–7.66)	9.5 (7.7–12.2)	9.7 (7.8–10.6)	0.605
Platelets, 10^9^/L	228 (209–228)	233 (189.5–296.5)	258.5 (218.5–341.3)	0.151
Erythrocyte, 10^12^/L	5.03 (4.74–5.03)	4.05 (3.57–4.91)	3.85 (3.4–4.6)	0.103
Hemoglobin, g/L	143 (134.25–143)	118 (102–138)	111.5 (98.8–133)	0.264
Creatinine, mmol/L	110 (96.25–110)	106 (93–122)	115 (98–137.8)	0.282
Stable angina grade II, %	65.2	64.7	65.3	0.590
Stable angina grade III, %	34.8	35.3	34.7	0.590
Diabetes mellitus, %	25.7	22.4	29.4	0.633
Myocardial infarction in anamnesis, %	63.6	61.2	70.6	0.608
Obesity, %	34.8	26.5	52.9	0.599

The data represent the medians and interquartile ranges (Me (C25—C75)). *p* < 0.05 between indicators of sASA and rASA patients (Mann-Whitney U test). To compare categorical variables, the χ2 test was used, except when expected frequencies in contingency tables were less than 5, in which case the Fisher exact test was used.

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
