# Peer review of "Resistance to Acetylsalicylic Acid in Patients with Coronary Heart Disease Is the Result of Metabolic Activity of Platelets"

_pharmaceuticals, 2020, doi:10.3390/ph13080178_

Round 1

Reviewer 1 Report

While this manuscript is interesting I believe it needs to be improved prior to acceptance for publication.

MAIN CONCERNS:

There is potential for a number of confounding factors to the results obtained that appear not to have been considered. Such as:

It is stated that the aspirin sensitivity was determined pre-CABG from bloods that were collected pre-op and that aspirin was ceased 5 days pre-op. Was the blood consistently collected from patients on a certain number of days after aspirin cessation? Can you be confident that pharmacokinetic/dynamics of aspirin between patients did not play a role in the sensitivity/resistance measures?  Were these aspirin sensitivity measures repeated post-op to ascertain whether the assignment of patients to the sensitive/resistant groups changed?

Why do you think CHD patients displayed a lower level of aggregation than controls? Were there other medications that may have impacted upon this outcome such as beta-blockers that are commonly prescribed for these patients and may not have been ceased pre-CABG.

What was the impact of the on-pump CABG versus off-pump CABG on the platelet parameters measured at Day 1? Did one group receive longer treatment or higher doses of heparin than the other? Heparin has been demonstrated to increase platelet aggregation to many inducers.

It is indicated that some patients received aspirin post-op while others received aspirin and clopidogrel. What was the effect of the combined  anti-platelet therapy on the platelet measures undertaken?

The Discussion section contains a lot of repeated description of the results. It would be preferable to start this section with a brief summary of the key findings then discuss each of these findings in the context of the published literature providing mechanistic insight. The manuscripts of Mateos-Cáceres (Thrombosis Haemostasis 2010) and Thomas (Scientific Reports 2014) should be included in the Discussion section and their results related to your findings.

MINOR ISSUES:

The Introduction to this manuscript was generally good, however, there was no explanation given as to why platelets were studied before and after CABG and the specific time frames studied (Day 1 & Day 8-10). What was the rationale for this study design? What was your hypothesis in relationship to the operative process?

Results section: It would help readers if you provide an explanatory sentence at the beginning of Section 2.2 as to what the lucigenin and luminol-enhanced assays are looking at. You do this in the Discussion and Methods sections, however, these come too late for comprehension of the results description.

Figure 1: The plots have become misaligned and the a-d lot assignment is not visible. I suggest you do not also use a, b, c to indicate the p-values between groups but use different symbols such as #, ∇ etc...

Could you please either indicate the 'n' values for each of the groups in the plots or use a column that represents the mean with each of the individuals values appearing as an overlaid dot - this would give an indication of the n number for each group.

Ensure the Y-axis is large enough for data to be seen (collagen plot)

As above with Figure 2.

Table 1: Please put the actual numbers of patients in for the gender row with the percentages in brackets. Please use a '.' not ',' for the p-value and decimal points.

How did you determine the dose of 3.36 mM ASA for the sensitivity assay? Was a dose titration undertaken?

Reviewer 2 Report

In their manuscript, Grinshtein et al., analyzed the activity of NAD(P)-dependent dehydrogenases sensitivity in platelets from Aspirin sensitive and resistant patients with coronary heart disease after coronary artery bypass grafting. The authors found that platelet aggregation activity was correlated with platelet Nox2 activity only in sASA patients and only before CABG. I found that manuscript is well written and experiments are well conducted. I have only some minor suggestions.

It could be better to present platelet aggregation values, not only for end point. Aggregation curves could be more appropriate.

It could be more informative to present the data as dot-plot, not as bar-graph.

In discussion, the authors could also discuss how their findings can be applied to other diseases, in particular diabetes.

Round 2

Reviewer 1 Report

Thank you for the changes made to the manuscript. 

This study involved studying platelet sensitivity to aspirin before and after CABG. Yet the Introduction still does not contain anything to suggest why this was done. The following is the only indication of why the study was done:

However, the role of metabolic processes (including synthesis of ROS) in the implementation of the functional activity of platelets in CHD patients with different sensitivity to ASA hasn’t yet been studied. Thus, the aim of our investigation was to study the metabolic activity of platelets in CHD patients responding and not responding to ASA with a decrease in aggregation.

You did the above, but you also chose to study platelets before and after CABG in the final sentences of the Introduction. Why?

You should indicate in the Methods that a titration was performed prior to the final selection of the aspirin dose used to determine platelet sensitivity.
